# A Review of Composite Phase Change Materials Based on Biomass Materials

**DOI:** 10.3390/polym14194089

**Published:** 2022-09-29

**Authors:** Qiang Zhang, Jing Liu, Jian Zhang, Lin Lin, Junyou Shi

**Affiliations:** 1Key Laboratory of Wooden Materials Science and Engineering of Jilin Province, Beihua University, Jilin 132013, China; 2College of Science, Beihua University, Jilin 132013, China; 3State Key Laboratory of Superhard Materials, Jilin University, Changchun 130012, China

**Keywords:** biomass, phase change materials, composite materials, preparation, multifunctional

## Abstract

Phase change materials (PCMs) can store/release heat from/to the external environment through their own phase change, which can reduce the imbalance between energy supply and demand and improve the effective utilization of energy. Biomass materials are abundant in reserves, from a wide range of sources, and most of them have a natural pore structure, which is a good carrier of phase change materials. Biomass-based composite phase change materials and their derived ones are superior to traditional phase change materials due to their ability to overcome the leakage of phase change materials during solid–liquid change. This paper reviews the basic properties, phase change characteristics, and binding methods of several phase change materials (polyethylene glycols, paraffins, and fatty acids) that are commonly compounded with biomass materials. On this basis, it summarizes the preparation methods of biomass-based composite phase change materials, including porous adsorption, microencapsulation based on biomass shell, and grafting by copolymerization and also analyzes the characteristics of each method. Finally, the paper introduces the latest research progress of multifunctional biomass-based composite phase change materials capable of energy storage and outlines the challenges and future research and development priorities in this field.

## 1. Introduction

With the rapid development of the economy, the dual pressures brought by energy depletion and environmental pollution have forced mankind to continuously explore new renewable and clean energy sources [1,2,3,4]. At the same time, the imbalance of energy supply and demand in space and time and low utilization efficiency further aggravate the waste of resources and environmental problems during energy development and utilization [5,6,7,8]. Therefore, researchers also turn their attention to developing energy utilization technologies while exploring new energy sources to improve the effective utilization of existing energy sources. Phase change materials (PCMs) can store/release heat from/to the external environment through phase change within a narrow temperature variation range, which can reduce the mismatch between energy supply and demand and improve the effective utilization of energy [9,10]. Starting to be studied in the 20th century, PCMs have been widely used in many fields, such as solar energy storage, industrial waste heat recovery, building energy saving, electronic components, and the thermal regulation of batteries [11,12,13,14]. However, traditional PCMs are prone to leakage during solid–liquid phase transition, which as an inherent defect that seriously hinders the large-scale application of PCMs [15,16].

Biomass materials are a class of renewable energy with extremely abundant sources in nature [17,18,19]. The development and utilization of biomass resources is of great significance for relieving the pressure on energy and the environment [20,21]. Most biomass materials have a natural pore structure, which can provide more adsorption sites to fix PCMs through interactions such as capillary force, surface tension, and van der Waals forces [22,23,24]. Biomass-based composite PCMs, i.e., the composites of biomass materials and PCMs, broaden the application range of biomass materials, realize their high-value utilization, and provide a new direction for the research and development of composite PCMs. At present, the application of biomass and its derived materials in the field of composite PCMs is still very limited. The development and utilization of diversified and multifunctional biomass-based composite PCMs has become the focus of future research.

This paper first summarizes the phase change characteristics and binding methods of PCMs commonly used to prepare biomass-based composite PCMs. Then, it introduces the preparation methods (porous adsorption, microencapsulation based on biomass shell, and grafting by copolymerization) of biomass-based composite PCMs and presents some multifunctional biomass-based composite PCMs. Finally, the paper provides an outlook on the application and characteristics of biomass materials in composite PCMs.

## 2. Phase Change Materials (PCMs)

PCMs can release or store heat through phase change at an almost constant temperature, which are a good choice for latent heat storage. Therefore, utilizing the characteristics of PCMs to store excess energy can effectively alleviate the problem of uneven energy distribution. PCMs fall into the categories of organic, inorganic, and eutectic ones, according to their chemical structures [25]. In particular, organic PCMs are most studied due to their high heat storage density, low undercooling, suitable phase change temperature, non-corrosivity, and stable properties. However, leakage during the solid–liquid phase change has limited the practical application of organic PCMs. This problem can be solved to a large extent by using biomass-based porous materials to adsorb PCMs (mainly including polyethylene glycols, paraffins, fatty acids) for preparing composite PCMs in stable shapes [26,27,28,29].

### 2.1. Polyethylene Glycols (PEG)

PEG is a class of typical PCMs which features high latent heat, suitable phase change temperature, and low thermal hysteresis [30,31]. Their controllable molecular weights enable them to have different properties at different average molecular weights. The properties of some common PEG are listed in Table 1. With the increase in molecular weight, the phase change temperature and latent heat of PEG also show an upward trend. In practice, PEG can be selected according to different application scenarios.

Jiang et al. [40] used PEG-10000 as PCMs and wood flour as support materials to develop new composite PCMs by direct impregnation (Figure 1). The results showed that the latent heat and phase change temperature of the composite were 90.9 J/g and 36.8 °C, respectively. The PEG was adsorbed on the wood flour by hydrogen bond interaction, capillary force, and surface tension. Leakage was not found during heating at 100 °C for 30 min. The composite materials showed excellent thermal properties and the ability of leakage protection. Moreover, the PEG can also improve the dimensional stability of wood. Li et al. [41] prepared a biomass-based composite PCMs by impregnating PEG-1000 into green fir wood and applying a varnish coating to prevent PEG from leakage. The composite PCMs had a phase change temperature of 26.74 °C and a latent heat of 73.59 J/g. After impregnation, the PEG penetrated the cell walls and then provided a thin, semipermeable layer to block the water molecules inside (Figure 2), which reduced the volume shrinkage of wood by 34.55% and improved its dimensional stability. In addition to wood, other biomass materials can also be combined with PEG to prepare composite PCMs. Zhang et al. [42] took nanosilver-coated eggplant-based biological porous carbon (BPC) as support materials and PEG-6000 as PCMs to prepare PEG-6000/BPC composite PCMs. The phase change temperature and the latent heat of PEG-6000/BPC were 59.8 °C and 147.8 J/g, respectively. PEG-6000 was adsorbed on the support materials by capillary force, and the resulting composite had a good anti-leakage ability. Wu et al. [43] selected PEG-4000 as PCMs and calcined diatom as support materials to prepare shape-stabilized PCMs (SSPCMs). The SSPCMs had a phase change temperature and a latent heat of 54.3 °C and 128.9 J/g, respectively. No leakage was observed when the SSPCMs were heated at 80 °C for 60 min.

In general, the biomass-based composite PCMs prepared by PEG have excellent thermal properties and leakage resistance, thus becoming promising materials in the fields of new environmental protection and energy-saving materials.

### 2.2. Paraffins

Paraffins are mainly straight-chain n-alkanes. They are characterized by high latent heat, no undercooling phenomenon, excellent thermal stability, and rich sources, which make them PCMs with immense potential [44]. The thermal properties of paraffins are related to the molecular chain length (the number of carbon atoms). Herein, Table 2 presents the properties of several common paraffins. Their phase change temperatures are found closer to the ambient temperature, and as a result of which they have been widely used for heat storage based on phase change.

Luo et al. [51] proposed a unique biomass-based composite PCMs formed by impregnation with a paraffin as the PCMs and garlic peel as the support materials. The composite underwent phase change at 60.2 °C and had a latent heat of 52.5 J/g. The paraffin was held in the pore structure by hydrogen bond interaction and van der Waals forces. The composite PCMs were heated at 80 °C for 60 min without leakage and showed good shape stability. Wang et al. [52] developed a new type of biomass-based composite PCMs by vacuum impregnation with the wild daisy stem carbonized at a high temperature as the support materials and a paraffin as the PCMs (Figure 3). The phase change temperature and the latent heat of the composite PCMs were 40.1 °C and 213.6 J/g, respectively. The pore structure of the wild daisy stem adsorbed the PCMs by capillary force. After heating at 70 °C for 2 h, the weight loss rate of the composite materials was only 2.1%, which indicated its good leak-proof performance. Yu et al. [53] used rice husk ash as support materials and paraffins as PCMs to prepare a biomass-based composite PCMs by impregnation. The composite PCMs had a phase change temperature of 48.2 °C and a latent heat of 95.7 J/g. After 300 cycles of heating and cooling, it still remained in excellent shape and had good thermal stability.

Currently, the composites of biomass materials and paraffins remain to be further studied. Especially, paraffin-based microcapsules are expected to form composites with biomass materials, which, however, is rendered difficult due to complex preparation and high costs.

### 2.3. Fatty Acids

Common PCMs of the fatty acid group include stearic acid (SA), lauric acid (LA), decanoic acid, myristic acid (MA), and palmitic acid (PA). Fatty acids have the characteristics of high latent heat, good thermal stability, no supercooling, and low costs [54,55].

Wen et al. [56] prepared composite PCMs by vacuum impregnation method with carbonized corn stover as the support materials and SA as the PCMs. Regarding the composite PCMs, phase change occurred at 67.62 °C, and the latent heat was 160.74 J/g. The physical interactions of capillary force and surface tension prevented the leakage of melt SA from the porous structure of carbonized corn stover. As displayed in Figure 4, the composite still had good shape stability after 3 h of heating at 80 °C. Due to the high phase change temperatures of single fatty acids (40–60 °C), different fatty acids are usually combined into binary or multiple co-melting systems to meet the requirements of temperature regulation and human comfort. Table 3 lists the phase change temperatures of some binary and ternary fatty acids after mixing. The phase change temperatures of the mixed systems can be determined by the Schroeder equation. Zhang et al. [57] used a eutectic mixture of LA-SA as PCMs and incorporated it into the carbonized corn cob to prepare SSPCMs (Figure 5). The resulting composite PCMs had a phase change temperature of 35.1 °C and latent heat of 148.3 J/g. Evidently, the phase change temperature of the binary fatty acid system was significantly lowered. LA-SA was fixed in the pore structure by physical adsorption mainly via surface tension and capillary force. After 200 thermal cycles, the composite PCMs can still maintain good shape stability. Sari et al. [58] designed leak-proof biomass-based composite PCMs with carbonized sugar beet pulp as the support materials and a eutectic mixture of CA-SA as the PCMs, which had a phase change temperature and a latent heat of 24.0 °C and 117.0 J/g, respectively. This phase change temperature was quite close to the human comfort temperature. The capillary force and surface tension between the support materials and PCMs can prevent the seepage of the molten PCMs. The latent heat capacity of the composite decreased by only 3% after 2000 cycles of cooling and heating.

However, fatty acids are mostly derived from animals and plants, and biomass materials are easily influenced by microorganisms. Therefore, it is necessary to further explore the biological durability of fatty acid–biomass composite PCMs.

## 3. Preparation Methods of Biomass-Based Composite PCMs

Biomass and its derivatives are commonly used as support materials to prepare composite PCMs because of their advantages of wide sources, abundant varieties, low costs, and simplicity to obtain and renew [69]. According to the research on biomass-based composite PCMs in recent years, their preparation methods can be roughly categorized as follows: (1) Natural biomass or its derived materials are employed as raw materials to develop porous adsorption matrices on which PCMs are then loaded so as to prepare biomass-based composite PCMs. (2) Biomass materials were used as shell materials to prepare microencapsulated PCMs (MEPCMs). (3) Biomass-grafted composite PCMs can be synthesized by copolymerization. The applicable methods for different types of PCMs and biomass materials and the specific conditions required for the preparation process are shown in Table 4.

### 3.1. Porous Adsorption

In the porous adsorption method, the unique pore structure of biomass materials is utilized to provide adsorption sites for the encapsulation of PCMs. After drying and high-temperature carbonization (400–1000 °C), biomass materials are used as the support materials, and then PCMs are introduced into the pore structure to prepare biomass-based composite PCMs. Alternatively, natural biomass-derived materials are used as raw materials to construct porous aerogel adsorption matrices with a three-dimensional structure, and then composite PCMs are prepared by the porous adsorption method.

Gu et al. [70] obtained biomass-based composite PCMs by direct impregnation with PA as the PCMs and biomass porous carbon prepared from pepper straw as the support materials. Since PA was successfully encapsulated in biomass porous carbon, the leakage problem was greatly improved. Zhao et al. [71] first prepared biomass hierarchical porous carbon through hydrothermal carbonation and high-temperature carbonation of water chestnut in the presence of aluminum hypophosphite as an activator and then biomass-based composite PCMs through vacuum impregnation by loading octadecane into the biomass hierarchical porous carbon (Figure 6). The maximum loading of octadecane can reach 85%, and no leakage occurred at that time. The phase change temperature and the latent heat of the composite PCMs were 34.2 °C and 216.2 J/g, respectively. Moreover, after 100 cycles of heating and cooling, its latent heat capacity decreased by merely 1%, indicative of good cycle stability. Polysaccharide biomass such as cellulose and its derivatives, lignin and chitosan, is often used to prepare biomass-derived aerogels on account of the good functionality, strong designability, and high added value endowed by the functional groups on its molecular structure. With cellulose as the raw materials and tert-butanol and deionized water as co-solvents, Wu et al. [72] prepared cellulose-based carbon aerogels through soaking expansion, orientation, freeze-drying, and high-temperature carbonization. Then, they loaded SA into the cellulose-based carbon aerogels by vacuum impregnation to form three-dimensional (3D) composite PCMs. The leakage rate of the composite PCMs was only 0.13% at 80 °C (higher than the melting point of SA), and its phase change temperature and latent heat were 67.92 °C and 201.53 J/g, respectively. In short, the composite PCMs exhibited excellent leak-proof performance and thermal storage capacity. After preparing biomass-based carbon aerogels with sunflower receptacle and stalk as raw materials, Wang et al. [73] realized 1-hexadecanamine/carbon aerogel composite PCMs by vacuum impregnation (Figure 7), which had a high heat storage capacity (above 200 J/g) and good shape stability. Wood is a kind of abundant biomass material in nature with a pore structure inside and is able to act as matrix materials to encapsulate PCMs. Ma et al. [74] prepared CA-PA/delignified wood composite PCMs by delignifying wood and loading CA-PA into the wood by vacuum impregnation. The obtained composite PCMs had a good encapsulation effect, whose phase change temperature and latent heat were 23.4 °C and 94.4 J/g, respectively.

In summary, the preparation of composite PCMs based on biomass materials has the advantages of simple operation, superior performance, and adjustable orientation. However, the types of biomass materials currently studied are far from sufficient. On the one hand, the scope of biomass materials should be actively developed, and on the other hand, new methods should be explored to prepare porous biomass matrices with better performance.

### 3.2. Microencapsulated Phase Change Materials (MEPCMs)

MEPCMs consist of PCMs core and a biomass shell. The biomass shell is capable of isolating the internal PCMs from the external environment, thereby preventing the internal PCMs from leakage at high temperatures. At present, there are not many biomass materials available as the shell materials for MEPCMs, mainly including starch, chitosan, and cellulose.

Songpon et al. [75] took a mixture of ethyl cellulose (EC) and methyl cellulose (MC) in a mass ratio of 2:1 as the shell materials and eicosane (C20) as the core materials to develop EC-MC/C20 microcapsules with simple self-assembly. The preparation mechanism is presented in Figure 8. The results showed that when the ratio of the core materials to the shell materials of the MEPCMs was 10:1, the latent heat of the microcapsule was as high as 239.8 J/g, higher than that of pure C20, which indicated an excellent heat storage capacity. This may be due to the interaction of C20 and the EC/MC polymeric shell materials interferes with the solidification of C20 inside the spheres in such a way that the transition is multi-stepped and more exothermic than that of the pure C20. Huo et al. [76] used a chitosan and styrene-maleic anhydride copolymer composite as shell and a comb-like polymer as core to achieve MEPCMs by coagulation. In addition, they investigated the effect of the core-to-shell mass ratio on the morphology and thermal properties of the microcapsules. The latent heat of the microcapsules was found to decrease gradually with the increase in the shell materials proportion. In the case of the core-to-shell ratio at 1:2, the microcapsules possessed a good microstructure and excellent heat storage and release capacity. Irani et al. [77] utilized graphene and starch as shell and n-heptadecane as core to prepare MEPCMs by self-assembly depending on the mechanism in Figure 9. The obtained microcapsules had a regular spherical morphology with a phase change temperature of 23.13 °C and a latent heat of 174.30 J/g.

At present, there are still few studies on the preparation of MEPCMs by coating PCMs with biomass materials. It is mainly because of the complex preparation process and high costs of microcapsules, which impede the development of MEPCMs. Therefore, it is necessary to develop new technologies, simplify preparation processes, reduce costs, and expand the application of biomass materials in MEPCMs.

### 3.3. Grafting by Copolymerization

Biomass-grafted composite PCMs adopt biomass materials as a fixed matrix on which the PCMs are grafted and fixed with the help of a cross-linking agent. This makes the molecular chain movement confined within the biomass network structure, thereby realizing the effective encapsulation of PCMs. Polysaccharide biomass materials are a good choice for the preparation of such composite PCMs because they contain a large number of hydroxyl groups on the molecular chain, which can provide active sites for grafting reactions [78,79].

In line with the polymerization mechanism in Figure 10, Yang et al. [80] prepared novel composite PCMs by solvent-free bulk polymerization using PEG as PCMs, diphenylmethane diisocyanate (MDI) as a coupling agent, and xylitol as a molecular framework. This composite PCMs can store or release heat through a solid–solid phase transition and avoid leakage during solid–liquid transformation. It had the phase change temperature and the latent heat of 41.65 °C and 76.37 J/g, respectively, and retains more than 40% of the heat storage capacity of pure PEG. Similarly, PEG can also be selected as the functional medium of phase change. Liu et al. [81] synthesized new solid–solid composite PCMs of polyurethane with castor oil as the molecular skeleton and MDI or hexamethylene diisocyanate (HDI) as a coupling agent. The composite PCMs prepared with HDI as the coupling agent had better heat storage and release ability. Its phase change temperature and latent heat were 51.4 °C and 117.7 J/g, respectively. Moreover, the composite PCMs was not subjected to any decomposition at 250 °C, which implied good thermal stability. Peng et al. [82] prepared four kinds of solid–solid PCMs capable of energy storage with different cross-linking densities by using PEG as the functional phase change segment and β-cyclodextrin as the molecular framework. They found that the cross-linking density had a significant effect on the energy storage capacity of the composite PCMs, which was inversely proportional to the heat storage performance. The composite PCMs with the lowest cross-linking density had the highest heat storage and release capacity. Its phase change temperature and latent heat were 60.5 °C and 114.8 J/g, respectively.

Intermolecular chemical reactions are the most effective encapsulation method for PCMs. However, overly efficient encapsulation restricts the molecular chain movement of the phase change segments, resulting in poor heat storage capacity of the composite PCMs. Moreover, grafting reactions have the disadvantages of complicated processes and high energy and time consumption. They also require a cross-linking agent to achieve cross-linking between molecules. Therefore, how to simplify the synthesis process, develop green organic synthesis technologies, and improve materials properties has become the future research focus for further development.

## 4. Multifunctional Biomass-Based Composite PCMs

At present, composite PCMs have been widely applied in energy-saving buildings, intelligent clothing, solar energy utilization, power batteries, electromagnetic radiation, chemical sensing, and many other fields. In the future, with the diversification of application scenarios, multifunctional composite PCMs will become an important research topic. Researchers have modified PCMs by physical or chemical means, and efforts are made to design and construct new composite PCMs with special functions such as photothermal conversion and thermochromism. Compared with traditional PCMs, functionalized composite PCMs further improve the comprehensive performance, expand the application scope, and create new ideas for the development of composite PCMs. However, intelligent multifunctional PCMs aimed at meeting greater demands for materials utilization remain to be developed.

### 4.1. Composite PCMs with Photothermal Conversion Ability

Solar energy is the most abundant and cleanest renewable energy in nature. Solar radiation can be converted into heat energy and stored in PCMs, which can bridge the intermittency and uncertainty of solar radiation to a large extent. Photothermal materials can be divided by type into carbon-based materials, organic materials, metal-based materials, semiconductor materials, and other photothermal materials. A good photothermal material should have a broad-spectrum absorption capacity. If photothermal materials and organic PCMs are combined, the excellent optical absorption ability of the photothermal materials can be leveraged to prepare composite PCMs with photothermal conversion and storage ability. This will be an advanced energy conversion and utilization technology with great potential and broad application prospects [83].

Hu et al. [84] synthesized novel composite PCMs having a photothermal conversion ability, with waste coffee grounds doped with 1–5wt% reduced graphene oxide taken as the support framework and a PEG as the functional medium for phase change. The results demonstrated that after being doped with reduced graphene oxide, the coffee grounds had a better 3D porous structure, which enhanced the uptake of PEG to 60.3%. The addition of reduced graphene oxide was highly conducive to the light absorption of the composite PCMs in the entire UV-Vis-NIR range, which started to store solar energy when the illumination time reached 245 s. Chen et al. [85] carbonized natural wood at a high temperature to produce biomass porous carbon. The porous matrix was then interacted with PCMs, n-octadecane, during vacuum impregnation to form biomass wood-based composite PCMs. Moreover, the surface layer of the composite PCMs was reinforced with a thin graphite layer, which further enhanced the stability of n-octadecane during phase change. As the matrix materials performed excellently in absorbing sunlight, the composite PCMs showed a photothermal conversion efficiency of up to 97%. In addition, the graphite layer enhanced the thermal conductivity of the composite materials and improved its heat storage and release rate, which was more conducive to its use in practical scenarios. Xie et al. [86] freeze-dried radishes and deposited a layer of polydopamine on their surface. A PEG as PCMs was combined with the matrix materials by dipping method with polydopamine as a binder to effectively prevent the leakage of PEG. In this way, a kind of sandwich-structured composite PCMs was synthesized (Figure 11). Thanks to the light absorption ability of polydopamine in a wide wavelength range, the composite PCMs realized photothermal conversion followed by energy storage.

The composite PCMs with photothermal conversion ability address the common problem of low solar energy utilization of PCMs and have the capacity of heat storage and release compared with photothermal materials. However, they are still in the laboratory stage, and no practical application has yet been reported. Currently, most of the research is centered on their preparation and performance while less on the comparison of specific PCMs, photothermal materials, and matrix materials. Additionally, the preparation process is complicated, and costs are high in general. According to the requirements of materials characteristics and application scenarios, selecting suitable photothermal materials and preparation methods for composite PCMs is still the research focus.

### 4.2. Thermochromic Composite PCMs

Thermochromic compounds generally consist of a coupler, a developer, and a co-solvent. Few studies have been reported on the visualization of the phase change process of PCMs with energy storage ability. Temperature visualization can be achieved if thermochromic materials are added to those PCMs. If the phase change process is visualized, the current ambient temperature can be easily determined without the aid of a thermometer, which greatly facilitates people’s lives.

Yang et al. [87] selected crystal violet lactone (CVL) and bisphenol A (BPA) as the color former and color developer, respectively, and mixed them with the PCMs 1-tetradecanol (TD) to make thermochromic PCMs. A novel thermochromic wood-based composite PCMs was prepared by loading the thermochromic PCMs into delignified wood by vacuum impregnation (Figure 12). Its phase change temperature and latent heat were 34.31 °C and 118.5 J/g, respectively. Moreover, the addition of thermochromic materials brought about an excellent reversible thermochromic ability of the composite PCMs, which can display the phase change process and temperature in real time through the color alteration from dark blue to off-white. Figure 13 shows the response of the composite PCMs to temperature change. Similarly, with CVL/BPA/TD as thermochromic PCMs and bamboo with lignin as support materials, Heng et al. [88] developed a new type of thermochromic bamboo-based composite PCMs by vacuum impregnation, which underwent phase change at 40.8 °C and had a latent heat of 113.3 J/g. In the range of 25–60 °C, the total color difference of the composite PCMs changed with temperature, and the discoloration temperature was about 40 °C, consistent with the DSC results. The bamboo-based composite PCMs exhibited excellent thermal properties and realized reversible thermochromism, thus having good application prospects in thermal insulation, temperature control, and especially interior decoration materials. Feng et al. [89] chose delignified bamboo as support materials and loaded the thermochromic compounds [3,3′-bis(1-n-octyl-2-methylindol-3-yl) phthalide (BP):BPA:TD] into the bamboo-based porous materials through the vacuum impregnation method to prepare a bamboo-based reversible thermochromic composite materials. They also investigated the effects of the BP-to-BPA ratio on the properties of reversible thermal discoloration and phase change energy storage. The experimental results showed that the synthesized bamboo-based reversible thermochromic composite PCMs had better thermochromic properties and thermal storage capacity when the thermochromic compounds were prepared at a mass ratio of 1:4:50 (BP:BPA:TD). The phase change temperature and the latent heat of the composite PCMs were 40.5 °C and 115.1 J/g, respectively.

The research on biomass-based thermochromic composite PCMs is still in its infancy, with few reports at present. The types of materials studied are still limited, and the preparation methods are relatively simple, namely the vacuum impregnation of porous materials in most cases. Therefore, researchers should strive to expand the scope of materials involved, develop new preparation techniques (such as for thermochromic MEPCMs) and enrich the forms and applications of thermochromic composite PCMs in future research.

### 4.3. Other Multifunctional Biomass-Based Composite PCMs

At present, multifunctional biomass-based composite PCMs still have great room for development. In addition to the abovementioned two categories of multifunctional biomass-based composite PCMs (photothermal conversion and thermochromism), other kinds have also attracted the attention of researchers, such as biomass-based composite PCMs with magnetothermal conversion, superhydrophobicity, and other functions.

After preparing carbon aerogel from kapok fiber, Song et al. [90] developed a kind of biomass-based composite PCMs by vacuum impregnation with carbon aerogel as the support materials and LA as the PCMs and then modified its internal microtubule structure with Fe_3_O_4_ nanoparticles (Figure 14). In this way, the composite PCMs performed well in magnetothermal conversion. The phase change temperature and the latent heat of the composite PCMs were 44.6 °C and 167.1 J/g, respectively. The heat storage capacity reached more than 95% of that of pure LA. In addition, Fe_3_O_4_ nanoparticles enabled the composite PCMs to have not only excellent magnetothermal conversion performance but also photothermal conversion ability and enhanced its thermal conductivity, playing the role of “one agent with multiple effects”. Hydrophobicity is also vital to biomass-based PCMs. If there is too much water, biomass materials are easily corroded by bacteria and fungi, which affects the use of biomass-based composite PCMs. Yang et al. [91] prepared wood-based composite PCMs by vacuum impregnation with delignified wood as the matrix materials and 1-tetraceol as the PCMs and sprayed a superhydrophobic coating on the surface to further strengthen the PCMs and make the composite materials hydrophobic. The results showed that the composite PCMs had a water contact angle of 155° due to the superhydrophobic coating. In the actual humid environment, the wood-based composite PCMs with the superhydrophobic coating performs better in heat storage and release. Hydrophobicity enables biomass-based composite PCMs to work in a more complex environment, further broadening their application scope.

At present, the research on multifunctional biomass-based composite PCMs has not formed a complete system, and plenty of fields remain to be explored, such as diversifying the preparation process and PCMs morphology (e.g., multifunctional energy storage MEPCMs), clarifying the impact of functional materials on composite PCMs while avoiding negative effects, and striving to achieve the goals of “synergistic enhancement” and “one agent with multiple effects”.

## 5. Conclusions

The combination of PCMs and biomass materials not only creates a new way for the reuse of biomass materials but also provides a new idea for the research and development of composite PCMs. This paper firstly introduces the basic properties, phase change characteristics, and combining methods of several phase change materials commonly used in composites with biomass materials, such as polyethylene glycols, paraffins, and fatty acids, and the preparation methods of biomass-based composite phase change materials are summarized, including porous adsorption, microencapsulation based on biomass shell, grafting by copolymerization, etc., and the characteristics of each method are analyzed. Finally, the latest research progress of multifunctional biomass-based composite phase change energy storage materials is introduced. Although the application of biomass and its derived materials in energy storage composite PCMs have made some achievements, it is still necessary to further broaden the research scope of these raw materials because they are abundant and actively explore the composite synergies of biomass materials and PCMs.

In the research and development of biomass-based composite PCMs, the priority is to tackle the leakage problem during the solid–liquid conversion of traditional PCMs. On this basis, it is necessary to fully exploit the potential properties of biomass and its derived materials and seek the best combination of them for new highlights. Future research should be concentrated on the following areas: exploring more biomass and its derived materials suitable for composite PCMs; finding new preparation methods for biomass-based composite PCMs, simplifying preparation processes, and reducing costs; and actively developing biomass-based composite PCMs that integrate multiple functions to make their application more comprehensive and high-value. This paper systematically compares and analyzes the methods for preparing biomass-based composite phase change materials, points out the advantages and disadvantages of each method, and provides theoretical guidance for the research and application of biomass-based composite phase change materials.

## Figures and Tables

**Figure 1 polymers-14-04089-f001:**
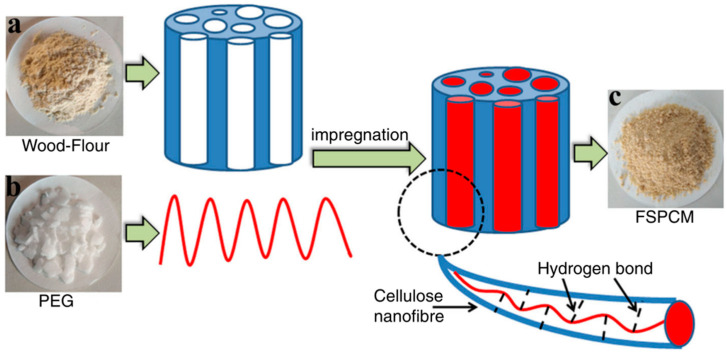
Preparation of composite PCMs with wood flour and PEG [40].

**Figure 2 polymers-14-04089-f002:**
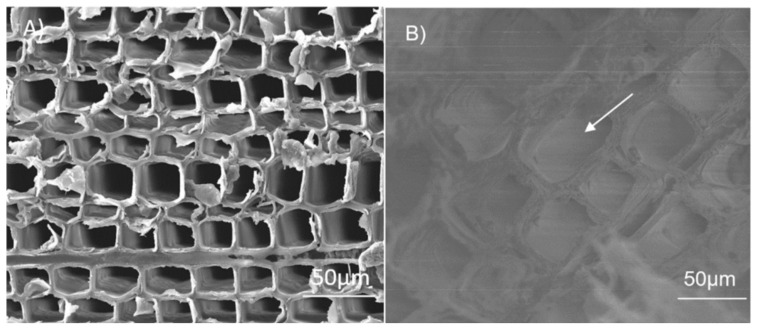
SEM images of the wood (**A**) before and (**B**) after PEG-1000 impregnation [41].

**Figure 3 polymers-14-04089-f003:**
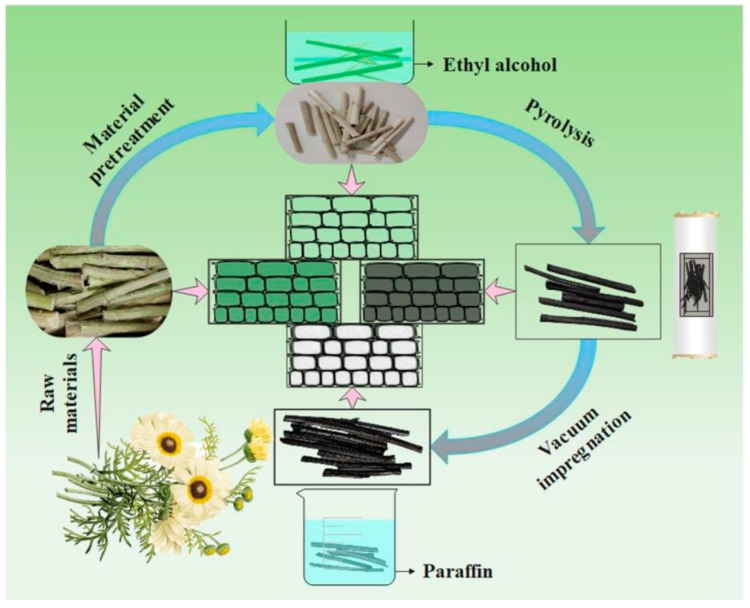
Preparation of composite PCMs with daisy stem and paraffin [52].

**Figure 4 polymers-14-04089-f004:**
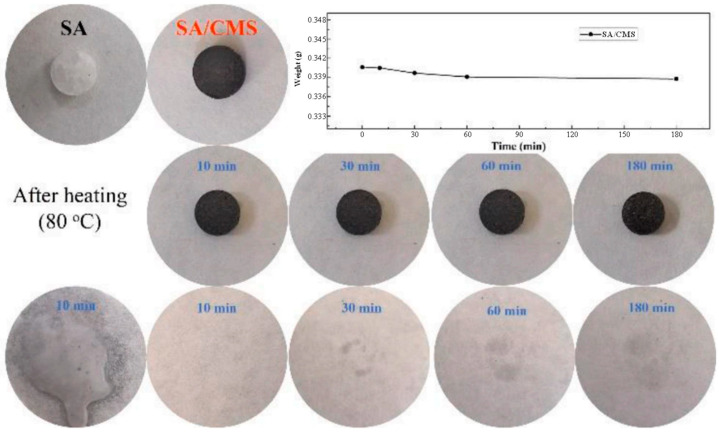
Leakage trace of SA and form and weight changes after different heating times [56].

**Figure 5 polymers-14-04089-f005:**
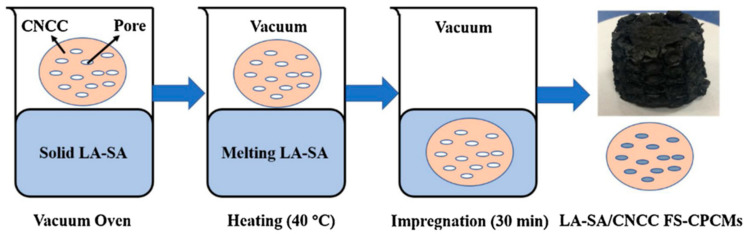
Preparation process of composite PCMs with carbonized corn cob and LA-SA [57].

**Figure 6 polymers-14-04089-f006:**
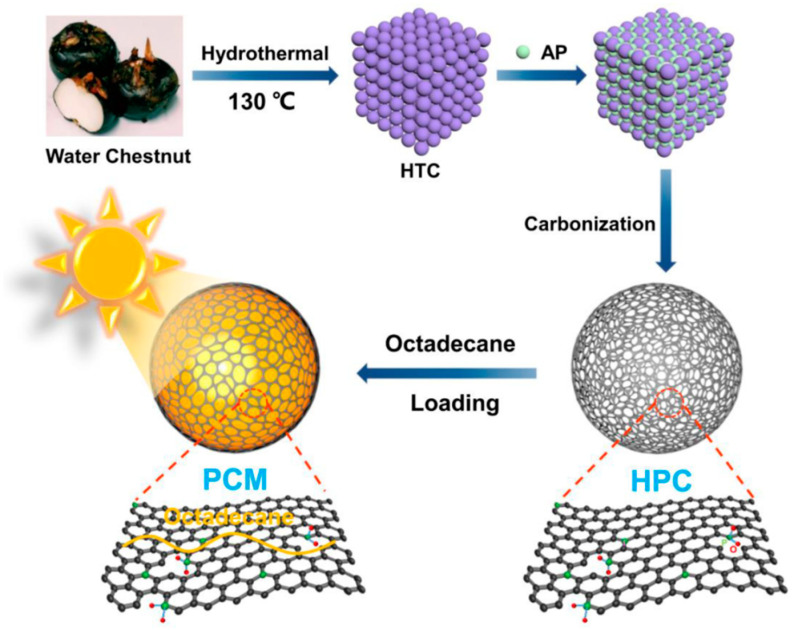
Preparation of composite PCMs with water chestnut and octadecane [71].

**Figure 7 polymers-14-04089-f007:**
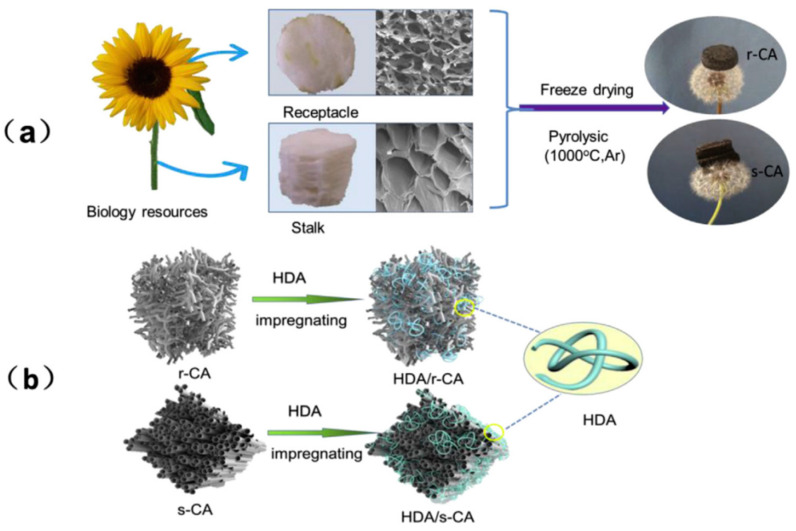
(**a**) Preparation routes of carbon aerogel, (**b**) Schematic illustration of 1-hexadecanamine/carbon aerogel composite PCMs [73].

**Figure 8 polymers-14-04089-f008:**
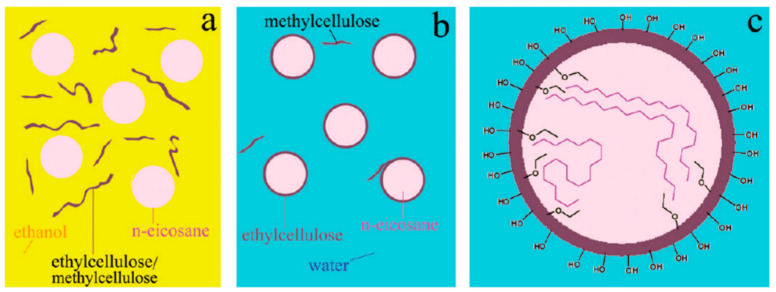
(**a**) Droplets of C20 were suspended in the ethanolic solution of EC and MC; (**b**) The water dispersible C20-encapsulated spheres; (**c**) EC-MC/C20 microcapsules [75].

**Figure 9 polymers-14-04089-f009:**
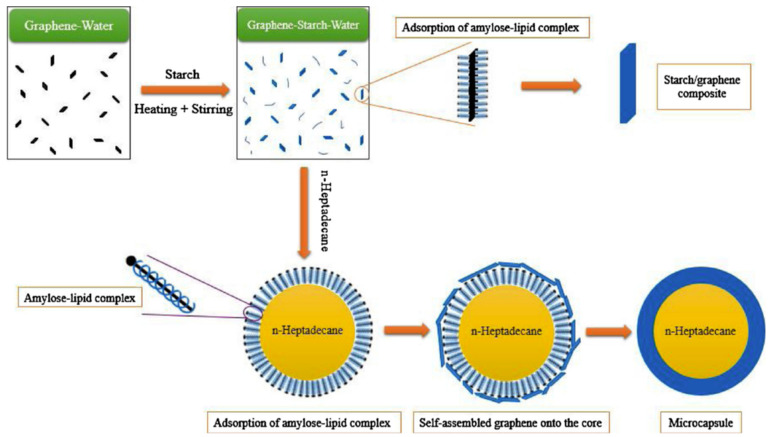
Preparation mechanism of starch/graphene shell microcapsules [77].

**Figure 10 polymers-14-04089-f010:**
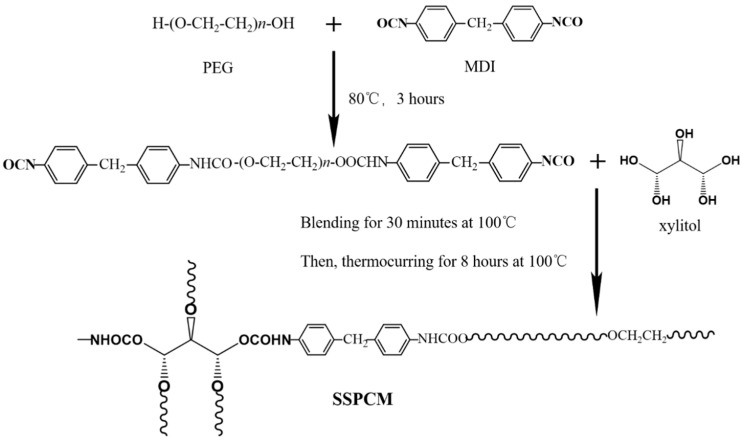
Diagram of polymerization mechanism [80].

**Figure 11 polymers-14-04089-f011:**
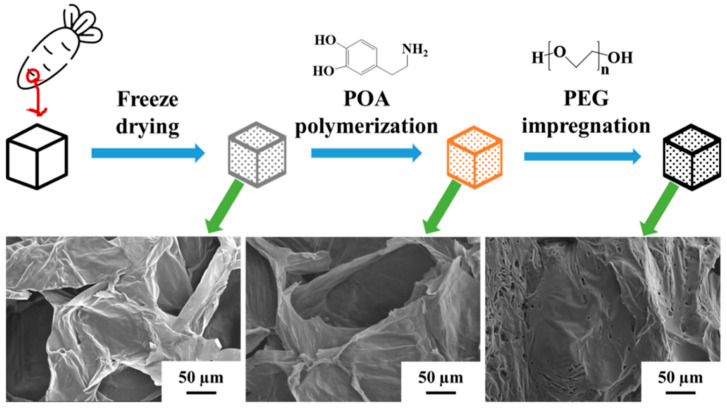
Preparation of composite PCMs with radishes and PEG [86].

**Figure 12 polymers-14-04089-f012:**
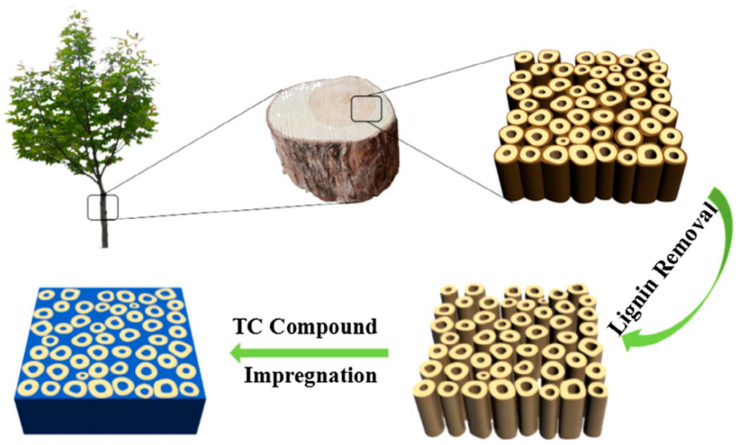
Preparation mechanism of thermochromic wood-based composite PCMs [87].

**Figure 13 polymers-14-04089-f013:**

Color changes of composite PCMs at different temperatures [87].

**Figure 14 polymers-14-04089-f014:**
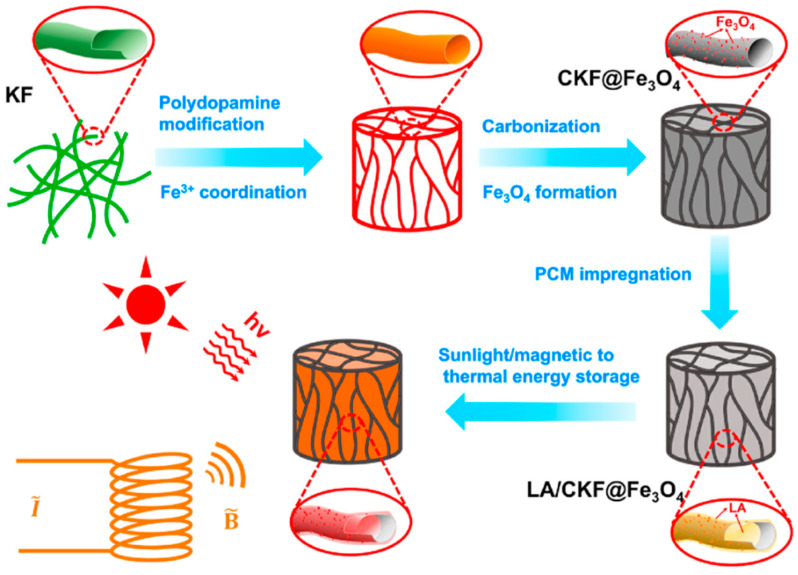
Preparation of composite PCMs with magnetothermal conversion ability [90].

**Table 1 polymers-14-04089-t001:** Phase change properties of PEG with different molecular weight.

Polyethylene Glycols (PEG)	Melting Process		Freezing Process		Reference
	T_m_ (°C)	ΔH_m_ (J/g)	T_f_ (°C)	ΔH_f_ (J/g)	
PEG-1000	42.8	129.3	23.6	129.8	[32]
PEG-2000	51.0	185.4	34.52	184.8	[33]
PEG-4000	60.5	172.4	41.96	207.0	[34]
PEG-6000	61.7	178.6	35.3	169.9	[35]
PEG-8000	64.6	180.0	44.3	167.9	[36]
PEG-10000	63.7	189.2	39.1	167.3	[37]
PEG-20000	67.7	160.2	42.9	155.7	[38]
PEG-35000	64.4	174.0	48.9	173.9	[39]

**Table 2 polymers-14-04089-t002:** Phase change properties of common paraffins.

Paraffins	Melting Process		Freezing Process		Reference
	T_m_ (°C)	ΔH_m_ (J/g)	T_f_ (°C)	ΔH_f_ (J/g)	
*n*-Dodecane (C12)	−6.1	219.0	−16.5	218.8	[45]
*n*-Tetradecane (C14)	5.88	225.8	2.15	225.4	[46]
*n*-Hexadecane (C16)	20.84	254.7	16.78	250.6	[47]
*n*-Octadecane (C18)	28.74	209.1	21.16	209.8	[48]
*n*-Eicosane (C20)	39.17	237.1	32.93	239.7	[49]
*n*-Docosane (C22)	42–46	234.4	36–39	233.6	[50]

**Table 3 polymers-14-04089-t003:** Phase change properties of different fatty acids and their complexes.

Fatty Acids	Attribute	Proportion	T_m_ (°C)	ΔH_m_ (J/g)	Reference
Lauric acid (LA)	Single	-	44.2	165.8	[59]
Myristic acid (MA)	Single	-	54.6	181.0	[60]
LA-MA	Binary	67.66:32.34	34.6	163.0	[61]
Palmitic acid (PA)	Single	-	62.8	207.0	[62]
Capric acid (CA)	Single	-	30.15	164.6	[63]
Stearic acid (SA)	Single	-	53.32	182.39	[64]
LA-SA	Binary	70:30	29.4	281.8	[65]
CA-MA	Binary	72:28	18.21	148.5	[66]
LA-PA	Binary	79:21	37.15	183.07	[67]
LA-PA-SA	Ternary	62.2:24.6:13.2	32.1	151.6	[68]

**Table 4 polymers-14-04089-t004:** The applicable methods for different types of PCMs and biomass materials and the specific conditions.

	Phase Change Materials (PCMs)	Specific Conditions	Biomass Materials
Porous adsorption	PEG, paraffins, fatty acids, etc.	High temperature carbonization	Most porous biomass materials
MEPCMs	PEG, paraffins, fatty acids, etc.	-	Starch, chitosan, cellulose, etc.
Grafting by copolymerization	PEG and others	-	Polysaccharide biomass materials

## Data Availability

Not applicable.

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
