# Peer review of "A Review of Composite Phase Change Materials Based on Biomass Materials"

_polymers, 2022, doi:10.3390/polym14194089_

Round 1

Reviewer 1 Report

The paper is well-written and comprehensive in a style that will be accessible to specialist and non-specialist readers.  I would suggest a few very minor corrections prior to acceptance for publication (see below).

(1) Lines 189-190. "high-temperature carbonisation".  Please give approximate temperature range.

(2) Lines 199.  "high-temperature carbonation".  Should this be "hydrothermal carbonation and high temperature carbonation" ?

(3) Lines 246-7.  I appreciate this is from the publication but can you comment on why the latent heat should be higher than in the pure PCM when there is mass dilution by the ethyl/methyl-cellulose shell and probably void spaces between the spherical microcapsules, so this is counter-intuitive?

(4) Line 447.  Suggest "synergies" or "benefits" not "law"

Author Response

Please check in attachment.

Reviewer 2 Report

Please find the review herewith.   1.      Pg. 7, line 196, what is PA?
2.      The nomenclature and abbreviation must be added in the table form.
3. The necessity of biomass-based BTMS is explained, but the manuscript could use more information and some quantitative data.
4. Three different preparation techniques are described, and the summary is also written well. However, which method must be selected for specific PCM, specific conditions, and specific bio-material can be a great addition to this section.
5. Since PCM stability is the most important factor, how about bio-PCM stability?
6. The manuscript must mention the drawbacks of biomass-based PCM.
7. The manuscript needs to explain how the review is novel.

Author Response

Please check in attachment

Reviewer 3 Report

Overall, this paper present a good finding that is worth for publication, however, some improvement is necessary before this paper is ready to be published.

Title:

-        The title need to be more specific since the word “Research progress” alone is not suitable to be read by author in the later year. It is suggested to use add the “Review” word to clearly indicate the type of this article, once it appear in the search results later.

Abstract:

-        The abstract should start with more comprehensive introduction of the interest of this paper.

-        “Biomass-based composite phase change materials and their derived ones are superior to traditional phase change materials due to their ability to overcome leakage.” What kind of leakage does the author refer to here? Please describe this in more accurate way.

-         

Introduction:

-        Citation should be placed before full stop, not after.

Section 3.2: MEPCMs should be written in full acronym first

-        Overall content: Author has written a good review on the topic provided. However, to make this review more interesting, more critical review can be highlighted for the findings reported from the point of view of the author.

Conclusion

-        In the conclusion, author first should summarize and conclude the overall review that has been made earlier. This then can be followed with the future direction of the study.

Author Response

Please check in attachment.
